# Multiparametric Comparison of Two TTA-Based Surgical Techniques in Dogs with Cranial Cruciate Ligament Tears

**DOI:** 10.3390/ani13223453

**Published:** 2023-11-09

**Authors:** Pedro Figueirinhas, José Manuel Gonzalo-Orden, Oliver Rodriguez, Marta Regueiro-Purriños, Ivan Prada, José Manuel Vilar, José Rodríguez-Altónaga

**Affiliations:** 1Departamento de Patología Animal, Universidad de Las Palmas de Gran Canaria, Trasmontaña S/N, 35416 Arucas, Spain; pedro.figueirinhas@fpct.ulpgc.es (P.F.); jose.vilar@ulpgc.es (J.M.V.); 2Departamento de Medicina y Cirugia Veterinaria, Campus de Vegazana, Universidad de León, 24071 León, Spain; jmgono@unileon.es (J.M.G.-O.); mregf@unileon.es (M.R.-P.); vetivi@hotmail.com (I.P.); jarodma@unileon.es (J.R.-A.)

**Keywords:** canine, lameness, cranial cruciate ligament tear surgery, tibial tuberosity advancement, implant, orthopedic, osteoarthritis

## Abstract

**Simple Summary:**

The rupture of the cranial cruciate ligament is one of the most common causes of hindlimb lameness in dogs. In this study, we compared two different tibial tuberosity advancement (TTA)-based surgical techniques to treat this condition. No significant differences were found when both procedures were analyzed using different assessment parameters.

**Abstract:**

Tearing of the cranial cruciate ligament causes hindlimb lameness in dogs. Different surgical procedures have been proposed to treat this condition. In this study, two different TTA-based techniques and implants were compared. A total of 30 dogs were separated into two groups according to the technique and implant used (Porous TTA^®^ or Model Xgen^®^). The aim of the study was to assess whether one of these techniques has better functional recovery of the joint, better bone consolidation after the osteotomy procedure and fewer osteoarthritic changes. We compared both groups up to 3 months after surgery. No significant differences were found in any of the assessed parameters. Thus, both procedures were found to be equally effective and safe.

## 1. Introduction

One of the main reasons dog owners visit a veterinary clinic is for lameness [1]. More specifically, the main etiology for hindlimb lameness are often conditions involving the stifle joint [2]. The cranial cruciate ligament (CCL) is the main structure that stabilizes the stifle, avoiding the cranial movement of the tibia and limiting the internal rotation and the hyperextension of the stifle [3]. Injury of the CCL alters these limitations and causes an abnormal movement between the stifle articular surfaces [4].

Repeated damage of the cartilage caused by interactions between the altered surfaces causes a progressive osteoarthritis (OA). Therefore, the goal of a CCL reconstruction is to restore the normal stifle dynamics in order to avoid the progression of OA [5,6].

Since the beginning of the 20th century there have been different approaches and discussions about the etiology, diagnosis, and treatments for a cranial cruciate ligament tear (CCLT) in the canine species [7]. The exact etiopathogenesis of the CCLT is unknown. Most of the diagnosed and treated cases are not preceded by trauma; instead, they are caused by a degenerative process that affects the collagen characteristics of the CCL, which can affect dogs of any gender or age [1].

The diagnosis of a CCLT can be achieved based on physical examination, imaging tests (X-ray, CT scan, MRI scan, ultrasounds) and arthroscopy [6]. CCLT treatment options can be medical, surgical or both. Regarding surgical techniques, different procedures have been proposed that aim to neutralize the cranial movement of the tibia by changing the tibia’s geometry. The axial force is redirected parallel to the patellar tendon and the tibio-femoral shear force is nullified, replacing the CCL function [8].

Tibial tuberosity advancement (TTA) techniques are based on an osteotomy that allows the advancement of the tibial tuberosity cranially. Then, it is stabilized using a special implant. This procedure was described in 2004 [9].

Since 2004, many authors have described other technical variations using different kinds of implants to obtain the cranial advancement of tibial tuberosity [10,11,12]. Among them, the Porous TTA^®^ by the Instituto Tecnológico de Canarias (Gran Canaria, Spain) and the Model Xgen^®^ by Securos Surgical with Xgen TTA (Securos Surgical, Fiskdale, MA, USA) techniques incorporate an osteotomy of the non-weight bearing portion of the tibia, where the implant is placed. The patellar ligament is aligned perpendicular to the common tangent of the femorotibial join, eliminating cranial tibial thrust. This new alignment eliminates the need for the CCL and results in a stable joint [9].

Porous TTA^®^ uses a porous wedge made of titanium that refills the defect made by the osteotomy. This wedge allows a fast and good vascularization that promotes abundant bone tissue formation. The fast penetration of bone into the implant enables the early stabilization of the stifle [13,14].

Model XGEN by Securos^®^ (Tibial Tuberosity Advancement System) is based on the original modified TTA technique by the Montavon and Kyon company in 2004 that was based on previous studies by Maquet [7]. This was the first system to use a forkless plate and cuttable cages. The cages are designed to establish early osteointegration (OI) and high resistance, reducing implant failure [10].

Several methods have been used to evaluate the hind function and OA progression after CCL treatment. In assessing stifle-related lameness, the Radiologic Bioarth Assessment Scale (RBAS) was first described in 2006 [15], and it is used to quantify the radiologic signs of OA in the joint of the stifle and classify the OA stage using a simple and objective method [15,16]. The method is based on a punctuation system that changes according to the radiologic changes observed on the OA of the stifle. To complement this, the Bioarth Functional Scale (FBAS) is a lameness scoring system (from 0 to 3 and 0 to 2 according to the case) that assesses 12 parameters, including functional limitation, joint mobility and muscular atrophy. Specifically, these parameters include changes in the limb while standing, changes in posture while getting up, lameness in rest, lameness after 10 min of walking, resistance to walk, resistance to run, climbing stairs, limitations doing small jumps (40–50 cm) and stifle mobility [17].

The aim of this study is to assess if one of these CCL techniques (the TTA Securos^®^ implant or the TTA Porous^®^ implant) has a better functional recuperation of the stifle, better bone healing after the osteotomy procedure and fewer osteoarthritic changes. Complications have also been reported; therefore, we also want to assess the safety of these techniques. 

## 2. Materials and Methods

### 2.1. Dog Selection and Groups

All the dog owners of this study were clients of the Veterinary Teaching Hospital of the University of Leon (Leon, Spain).

A total of 30 dogs with unilateral CCLTs were used. Body weight and age ranged between 5 and 65 kg and between 12 and 210 months, respectively. The animals were randomly distributed into two groups by gender, breed and affected side (Appendix A).

Group 1—TTA with the implant Securos^®^ (model XGEN).

Group 2—TTA with the implant Porous^®^.

The inclusion criteria comprised the absence of any concurrent systemic or orthopedic diseases, and these criteria were assessed through hematologic, blood, and urine biochemical profiles. Furthermore, the subjects in the study could not have received any form of treatment for a minimum of one month.

A comprehensive clinical evaluation encompassing physical, neurologic and orthopedic examinations and assessment of vital signs was conducted to ensure that the sole cause of lameness in the subjects was specific joint osteoarthritis.

The physical examination was based on a complete evaluation of the affected limb, including pain and inflammation, and observing the dog standing and walking at different speeds (Appendix A). The functional examination of the limb was performed using a procedure based on the FBAS (Appendix A).

For hind limb examination, sedation with dexmedetomidine (3–5 μg/kg IM) and butorphanol (0.1 μg/kg IM) was used in order to perform the drawer test and the Finochietto test, also known as the jump test (an orthopedic test to assess meniscus tears in the stifle) [18].

### 2.2. Surgical Procedures

In Group 1, the Model XGEN by Securos^®^ implant was used; the Porous^®^ TTA implant was used in Group 2.

The surgeries were performed by two experienced surgeons. As we had two groups of fifteen animals, one surgeon performed seven surgeries using Porous TTA and eight using Model XGEN, whereas the other surgeon did eight surgeries using Porous TTA and seven using Model XGEN.

Both techniques have been previously described, but briefly:

Group 1: TTA with the Securos implant.

The patient was positioned in lateral recumbence so that the affected limb is lying flat on the table. The medial aspect of the proximal tibia is approached. A cranio-medial parapatellar incision is made from the patella to the medial saphenous vein.

The fascia is sharply incised and elevated until the tuberosity is medially and laterally visible. The patellar ligament is isolated by making an incision caudally and a small Gelpi retractor is used to protect it from the saw blade when performing the osteotomy.

The osteotomy line begins at the distal aspect of the tibial tuberosity and ends at the caudal arm of the Gelpi. The osteotomy must be curved gently in the distal tuberosity to minimize stress and avoid postoperative tibial fractures. Once the osteotomy is performed, the pre-contoured plate is placed with its cranial aspect parallel to the tuberosity and the screws are placed behind the cranial cortex. The distal screw should be positioned just above the osteotomy. Once the osteotomy cut is complete, the osteotomized tuberosity is spread apart to facilitate the TTA cage placement. Cancellous bone grafts can be collected from the tibial shaft and be placed later in the completed TTA cage. The width of the TTA cage was determined based on preoperative radiographs.

Finally, a 2.0 mm drill bit is used to fix the distal plate to the tibial shaft. The previously collected cancellous bone is now placed in the osteotomy gap (Figure 1a,b) [9].

Group 2: TTA with the Porous^®^ implant.

Before starting the surgery, the advancement of the tibial tuberosity should be measured.

The patient is positioned in lateral recumbence with the damaged stifle directly on the table. A medio-proximal approach to the tibia is performed. The incision starts over the patella and runs 1 cm distally to the end of the tibial tuberosity, and the proximal tibia is exposed. After drilling a distraction hole, an osteotomy is performed using a cutting guide. Then, the tibial tuberosity is slowly advanced distracting 1 mm per minute to avoid breakage. After this, the appropriate porous titanium wedge is placed to avoid cranial thrust. The wedge should be placed a few millimeters distally to the proximal aspect of the tuberosity. Finally, the plate is placed, fixing the tuberosity to the tibial diaphysis (Figure 2a,b) [19].

### 2.3. Postoperative Assessment

For OA assessment, radiographs were taken for all dogs on the day of surgery and one month and three months after surgery, with mediolateral and caudocranial projections. In the mediolateral projection, we assessed and scored the lips of the trochlea, proximal and distal poles of the patella, femoral condyles, tibial tuberosity, sesamoids bones of the gastrocnemius muscle and tibial plateau or proximal articular surface of the tibia. In the craniocaudal projection, we evaluated and scored the tibial plateau or articular surface proximal tibia, lateral epicondyle, medial epicondyle, intercondylar fossa of femur, head of the fibula and edge of the medial condyle of the tibia. The scoring values ranged from 0 to 3 points depending on the radiological signs of OA (0 points for no radiological signs of osteoarthritis, 1 point for mild osteoarthritis, 2 points for moderate osteoarthritis and 3 points for severe osteoarthritis). To determine the degree of total OA, the sum of the assigned score was be added to each of the anatomical indicated areas. This radiological assessment was performed by two veterinarians in each clinical case and each assessment score was averaged. Both veterinarians were experienced orthopedic surgeons. The values obtained by the two veterinarians were added to the final results table, placing the stifle in one of the following four groups: 0–2 points for stifles without radiological signs of osteoarthritis, 3–8 points for stifles with mild osteoarthritis, 9–18 points for stifles with moderate osteoarthritis and ≥18 points for stifles with severe osteoarthritis (Appendix A).

Osteointegration (OI) was evaluated according to an increase in the width and the extent of radiolucent lines between the bone and implant over time in the patient. These could be a sign of a lack of OI. The formation of radiopaque lines at the points of osteotomy indicates proper OI [20]. In our study, the presence of radiopaque lines at the points of the osteotomy was considered as good OI (Appendix A).

### 2.4. Experimental Design

The dogs were randomly assigned to each of the experimental groups. The groups were initially tested for comparability (Appendix A); there were no significant differences between the groups in any relevant variables before the treatments were applied. The variables used to assess the effect of each surgery were measured on an ordinal scale, with values ranging from 0 (good condition) to 3 (worst condition) and were summarized as median and quartiles. A comparison of the distributions of these variables between the two treatments was performed using the Wilcoxon–Mann–Whitney test for independent samples. To compare each treatment’s effects before and after surgery, the Wilcoxon test for paired samples was used. The proportions (of gender, side of disease, pain, etc.) between groups were tested using the chi-squared test. McNemar’s test was used to compare the proportion of dogs with good OI at one and three months after surgery.

We also calculated the length of time of each surgery for each dog (Appendix A).

In all cases, a significance level of 0.05 was used.

## 3. Results

The comparison data of the characteristics and clinical findings of dogs in Groups 1 and 2 prior to the surgery is summarized in Table 1. Considering the *p*-values, there were no significant differences between the groups in any of the variables considered.

The Bioarth Functional Scale scores prior to surgery are shown in Table 2. No significant differences were found between the groups in any of the variables considered.

The results three months after the surgical procedure were also compared in Table 3. Considering the *p*-values, there were no significant differences between both TTA techniques for any of the variables considered.

Raw Bioarth scale data can be found in the Appendix A).

Radiologically, there were almost no changes in OA during the three months after surgery. There were only two dogs (#1 and #22) that slightly increased their score (Appendix A). Regarding the OI of the implant in the first and third months after surgery (Appendix A), there were no significant differences between the two techniques.

Regarding post-surgical complications, only one minor complication was present in one dog from Group 1 (superficial wound) and there was one major complication in each group (implant failure and avulsion of tibial crest, respectively) (Appendix A).

Both procedures took a similar length of time to be performed (43.26 vs. 43.60 min) (Appendix A).

## 4. Discussion

Surgical joint stabilization to avoid OA progression in the CCLT and the restoration of limb function is still challenging for clinicians and researchers. Until now, none of the surgical procedures have been shown to completely fulfill these aspects [21]. In the present study, we compared the effectiveness of two different TTA-based surgical techniques to treat CCLTs.

Vezzoni et al. showed that 71% of CCL injuries are degenerative and 29% are caused by trauma; in both cases, the tear was complete [22]. In our study, we observed that 13% of CCL injuries were caused by trauma and 87% were degenerative.

The percentage of dogs with a CCLT in our study match with the values from Johnson et al. (1.55%), Whitehair et al. (1.82%) and Witsberger et al. (2.55%) [23,24,25]. We found 45 cases of CCLT in a total of 2400 dogs, a percentage of 1.87%, which is similar to the above studies.

The efficacy of TTA has been widely proven since 2004 when the technique was first described [9]. Many years later, TTA is considered a successful surgery to treat CCLT [10].

There are many different parameters we may use to compare the effectiveness of surgical techniques to treat CCLTs. One study used a pressure platform analysis, which is an objective parameter, performed prior to surgery and at four different postoperative time points to obtain a short-term comparison of tibial tuberosity advancement and tibial plateau levelling osteotomy (TPLO) [26]. Livet et al. used radiographic examination, lameness score evaluation and gait analysis to compare outcomes associated with TPLO and TTA [27]. These parameters are subjective but in very relevant. Another study critically reviewed the available literature focused on the preoperative planning, surgical procedure, follow-up and complications of CCLTs using different tibial tuberosity advancement techniques [28]. This study concluded that nearly 90% of the stifles examined in short-, mid- and long-term follow-ups showed full and acceptable functionality. It did not find any significant differences between TTA techniques. In our opinion, in order to compare very similar techniques, the validation of any results should be obtained from homogeneous groups. For this reason, we compared multiple characteristics of the dogs from both groups (gender, lameness in the affected limb, proportion of dogs with pain, etc.), finding no significant differences between the groups.

Regarding the techniques compared in our study, one study demonstrated the effectiveness of the Porous TTA in 61 dogs, which had a minor complication rate of 47.69% after 3 weeks, 10.77% after 6 weeks and 4% after 12 weeks of surgical intervention [13].

The Securos TTA is similar to the modified Maquet TTA; however, in this case, the tibial tuberosity is completely cut, leaving the Maquet hole apart. It has been concluded that the modified Maquet TTA obtains similar outcomes and complication rates when compared with traditional TTA [10]. In our case, the comparison of initial values between groups for all the physical variables of the animals showed no significant differences between groups, as all the *p*-values were greater than 0.05.

Regarding the evolution of OA, one study compared the OA changes in 33 stifles from 24 dogs treated with TTA and TPLO [29]. They concluded that the OA had progressed a little bit more in TTAs than in TPLOs but that the difference was not significant. In our study, only two dogs from the first group showed a minimal progression of OA, allowing us to conclude that there were also no significant differences between the TTA techniques.

Concerning the OI of the implants, we also obtained very good results for both groups after 1 month (86.7%) and after 3 months (93.3%). This fact is in concordance with a previous study, proving that porous TTA implants show excellent OI and osteoconduction properties [30].

In our study, we also compared the presence of complications when using both techniques. A previous study by Matchwick et al. reported a complications rate of 15.2% in TTAs performed by six different non-specialized surgeons, where 7.5% were major and 7.7% were minor complications [31]. Costa et al. obtained a complications rate of 13.4%, where 1% had implant failure, 1.2% had patella luxation and 0.9% had tibial tuberosity fractures [32]. They also concluded that complications are uncommon when performing TTAs. In our case, we believe that both major complications were due to the same problem: on the first surgeries using the Porous implant procedure, the osteotomy was made too close to the external cortical of the tibia. We realized this caused the avulsion of the tibia. After correcting this issue, the problem stopped occurring for the subsequent surgeries.

Finally, we compared surgery times in both groups, and we can confirm that there are no significant differences between them.

Although we tried to provide a sound study design, our study has some limitations. The first limitation is that the study only assesses a three-month evolution and problems affecting implant stability and/or OA progression may arise over a longer period of time. Second, the assessment of the functional status prior to and after surgery should have improved with the objective methods based on kinetic and kinematic parameters, as shown by other authors [32]. However, this was not possible because these methods are limited to animals of a certain size and our group was very heterogeneous and included patients of small sizes. Lastly, a larger number of patients in our study would have had given greater statistical significance. However, it was challenging to schedule the necessary reviews and reevaluations to accommodate the patients’ owners.

## 5. Conclusions

After comparing both groups 3 months post-surgery, we can conclude that the functional recuperation is similar for both procedures. This study also shows that the OI of the implants after 3 months is correct in both procedures. We also observed no progression of OA in both groups; therefore, we believe that there should be future radiologic and clinical control assessments to determine the long-term effects of these procedures. Minimal complications were noticed; therefore, we can confirm that both procedures are safe in a short-term follow-up.

## Figures and Tables

**Figure 1 animals-13-03453-f001:**
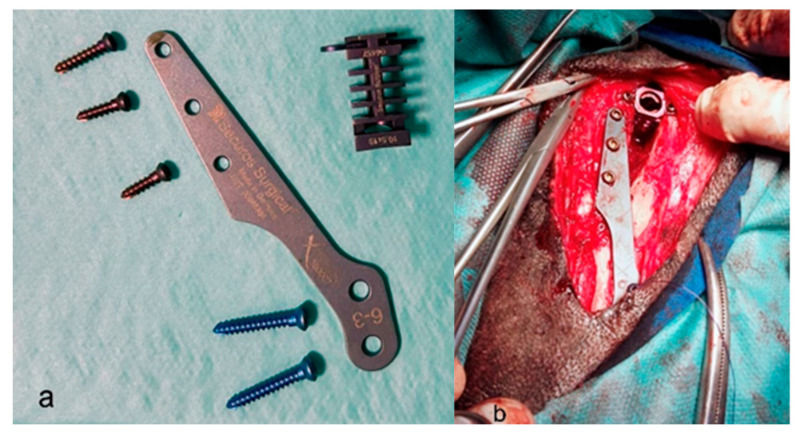
(**a**) View of the Securos implant. (**b**) View of the placed implant during the surgical procedure.

**Figure 2 animals-13-03453-f002:**
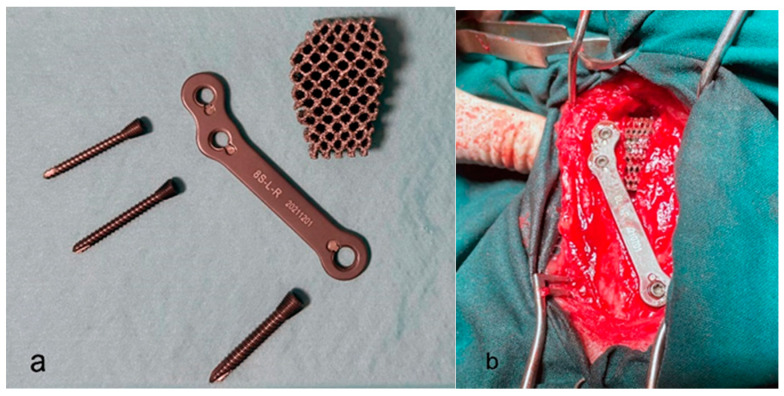
(**a**) View of the Porous implant. (**b**) View of the placed implant during the surgical procedure.

**Table 1 animals-13-03453-t001:** Comparison of gender, side of the disease, and physical examination of dogs with CCLTs treated with two different TTA-based techniques.

Variable	Group 1 (*n* = 15)	Group 2 (*n* = 15)	*p* Value
Female	8 (53.3%)	7 (46.7%)	1.0000
Spayed	3 (20.0%)	5 (33.3%)	0.6800
Right stifle	6 (40.0%)	8 (53.3%)	0.7140
Pain	12 (80.0%)	11 (73.3%)	1.0000
Inflammation	13 (86.7%)	14 (93.3%)	1.0000
Joint leak	13 (86.7%)	12 (80.0%)	1.0000
Finochietto	5 (35.7%)	6 (37.5%)	1.0000
Drawer test			0.8560
Absent	3 (20.0%)	2 (13.3%)	
Present	8 (53.3%)	8 (53.3%)	
Clear	4 (26.7%)	5 (33.3%)	

**Table 2 animals-13-03453-t002:** Mean and standard deviations of different variables using the Bioarth scale at baseline in each group.

Variable	Group 1	Group 2	*p*-Value
Changes in the affected limb while standing	1.60 ± 0.74	1.73 ± 0.70	0.5700
Changes in posture while getting up	1.00 ± 0.65	1.20 ± 0.68	0.4157
Lameness	2.13 ± 0.99	2.07 ± 0.96	0.8538
Lameness after 10 min of walk	1.67 ± 0.90	1.53 ± 0.99	0.7103
Resistance to walk	0.87 ± 0.92	0.80 ± 0.94	0.8065
Resistance to run and play	2.00 ± 0.53	1.60 ± 0.91	0.1363
Resistance to climb stairs	1.20 ± 0.94	1.40 ± 1.06	0.6115
Limitation to take small jumps	1.07 ± 0.70	1.07 ± 0.80	1.0000
Manual articular mobility of the stifle	1.33 ± 0.72	1.40 ± 0.74	0.7797
Limitation of the articular flexion movement	0.87 ± 0.52	0.80 ± 0.56	0.7370
Limitation of the articular extension movement	0.87 ± 0.52	0.80 ± 0.56	0.7370
Muscular atrophy	0.60 ± 0.63	0.53 ± 0.52	0.8690

**Table 3 animals-13-03453-t003:** Mean and standard deviations of different variables using the Bioarth scale three months after surgery in each group.

Variable	Group 1	Group 2	*p*-Value
Changes in the affected limb while standing	0.33 ± 0.49	0.20 ± 0.41	0.4326
Changes in posture while getting up	0.47 ± 0.52	0.53 ± 0.64	0.8871
Lameness	0.73 ± 1.03	0.80 ± 1.01	0.7840
Lameness after 10 min of walk	0.33 ± 0.82	0.33 ± 0.49	0.5182
Resistance to walk	0.40 ± 0.91	0.13 ± 0.52	0.3087
Resistance to run and play	0.33 ± 0.62	0.33 ± 0.49	0.8153
Resistance to climb stairs	0.27 ± 0.46	0.13 ± 0.35	0.3856
Limitation to take small jumps	0.00 ± 0.00	0.13 ± 0.35	0.1641
Manual articular mobility of the stifle	0.27 ± 0.59	0.13 ± 0.52	0.3431
Limitation of the articular flexion movement	0.13 ± 0.35	0.13 ± 0.35	1.0000
Limitation of the articular extension movement	0.07 ± 0.26	0.07 ± 0.26	1.0000
Muscular atrophy	0.33 ± 0.62	0.13 ± 0.35	0.3554

## Data Availability

Data are contained within the article and Appendix A.

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
