# Peer review of "Multiparametric Comparison of Two TTA-Based Surgical Techniques in Dogs with Cranial Cruciate Ligament Tears"

_animals, 2023, doi:10.3390/ani13223453_

Round 1

Reviewer 1 Report

Comments and Suggestions for Authors

The study “Multiparametric Comparison of two TTA-based Surgical 2 Techniques in dogs with Cranial Cruciate Ligament Tearcompared the Porous TTA ® and Model Xgen ® and concluded that both implants were equally effective and safe.

In general, the paper is easy to read. Unfortunately, I have some concerns with the quality of some content within the manuscript.

The English quality of the manuscript must be improved.

Details of the text are provided in additional comments (PDF).

Introduction

I suggest that the authors change the introduction with a focus on treatments of the cranial cruciate ligament tear. Statements such as lines 34-18 did not contribute to the focus of the manuscript.

Maybe you could start from line 44. Since the beginning....

Also, the paragraph should be contain more than one sentence.

Thus, you could start in line 44 and finish in line 54 (…and arthroscopy).

Also, the introduction is not the place to include figures. The figures can be included in the Material and Methods. You must include figures so that the reader can see the implants. For example, you could take a picture of the implant (a) and include together that the picture you presented in the paper (b).

You can improve the quality of the pictures by showing less of the surgical drapes.

You must include the author in each sentence. The reader can not know who wrote the statement if you do not include the author.

I think you must include a sentence before “the stifle-related lameness…” to make a link. Something such as: “Several methods have been used to evaluate the hind limb function and osteoarthritis progression after cranial cruciate ligament rupture treatment.”

It is just an idea. You can improve the sentence.

Material and Methods

Please, include the criteria of inclusion and exclusion used to select the dogs.

Finochietto test – Please explain.

Please explain the veterinarians who evaluated the X-rays. Are they radiologists or surgeons? Are they experienced veterinarians?

Were the surgeries made by one or more surgeons?

How did you evaluate the osteointegration?

Results

The title of the tables must better explain the variables that are evaluated.

Example:

Something like this:

Table 1. Comparison of gender, side of the disease, and physical examination in dogs with cranial cruciate ligament rupture treated with two different TTA-based techniques.

Discussion

There are several data that were discussed, but they are not in the results. Please also include in the results.

Line - lameness in the left paw. Please explain. I could not find about the left paw reading the manuscript.

Please include the limitations of the study.

Conclusions

Minimum complications were noticed; therefore, we can affirm that both procedures are safe.

Maybe you should include that the procedures are safe in short-term follow-up.

References

Please check and standardize.

I made some observations.

Supplemental material

In general, tables do not have grid lines. Please correct.

Table S1 – Spayed for females

Neutered for males.

Table S3. – Stifle better than knee

Table S5 – Hind limb

Table S7 - at the beginning

Mean score – period – not comma. Example: 0,3 – 0.3 

Comments on the Quality of English Language

The English quality of the manuscript must be improved.

Author Response

Please se the attachment.

Reviewer 2 Report

Comments and Suggestions for Authors

The work is original, but requires changes to improve the description of the methodology and results:

Line 27: Remove "," at the end of the sentence.

Line 67: Replace “ITC, Gran Canaria” with “Instituto Tecnológico de Canarias (Gran Canaria, Spain)” and “Model Xgen ® by Securos Surgical” with “XGEN TTA (Securos Surgical, Fiskdale, Massachusetts, USA)”

Line 75: Modify the entire introduction of the two techniques and put appropriate bibliographic citations.

Lines 75-88: The statement "The Model XGEN by securos® (Tibial Tuberosity Advancement 82 System) is based on the original modified TTA technique by Maquet 83 and increases surgical options while reducing surgical inventory " is incorrect. The two techniques are modifications of the technique developed by Montavon and the Kyon company, based on previous studies by Maquet. It is these authors who introduced the technique in veterinary medicine. Both Porous TTA and Xgen TTA are modifications of the technique described in 2004.

Line 75-88: Provide a brief description of the significant differences between the two surgical techniques. Please, briefly describe the two surgical techniques. Many readers of the paper may not be familiar with the surgical procedure.

Line 83 There are better bibliographical citations to indicate the origin of the TTA surgical technique. TTA was developed by Montavon and the company KYON.Line 80: "Enhance the image of the figure 1: Place a photo perpendicular, without the surgical field and surgeon's gloves predominating."

Line 88: Enhance the image of the figure 2: Place a photo perpendicular, without the surgical field and surgeon's gloves predominating.

Lines 106,113 and 114: “Securos® or Porous®” are not surgical techniques. Please define precisely.

Line 128: Why do radiological check-ups take x-rays in the "mediolateral and craniocaudal" projection and not in the "mediolateral and caudocranial" projection? Caudocranial projections provide a more realistic radiographic image of the implant's position, and no deformations occur."

Line 153: In “Experimental Design”, please review the numbering of all the tables. Errors have been detected.

Line 180: Replace “The Biopath functional scale score prior to surgery is shown in (table 2):” with “The Bioarth functional scale score prior to surgery is shown in the table 2.”

Line 173: Replace “The comparison data of characteristics and clinical findings of dogs in groups 1 and 2 prior to the surgery is summarized in (Table 1)” with “The comparison data of characteristics and clinical findings of dogs in groups 1 and 2 prior to the surgery is summarized in the table 1.”

Line 188: Replace “The results three months after the surgical procedure were also 188 compared (table 3)” with “The results three months after the surgical procedure were also compared in the table 3.”

Lines 215, 227, 231, 273 and others. Put the name of the authors, instead of “a study”.

Line 313: In “References”, Review all the bibliography and correct its presentation.

Line 318: Is the bibliography citation correct?

Line 346: The bibliographical citation is poorly referenced. AVEPA is not a paper.

Line 352: Put the original bibliographical citation, not the edition translated into Spanish.

Reviewer 3 Report

Comments and Suggestions for Authors

Comments on the Quality of English Language

I highly recommend the English to be reviewed by an English native speaker. 

Round 2

Reviewer 1 Report

Comments and Suggestions for Authors

I would like to thank the authors for submitting the revised manuscript. This resubmission is a significant improvement from the first submission. I have a few suggestions.

1. Introduction

Line 71 - The Porous TTA® uses...

Suggestion: Include this part together the next paragraph (The Model XGEN…)

The Porous TTA® uses a porous wedge made of titanium 72 that refills the defect made by the ……The Model XGEN by Securos® (Tibial Tuberosity Advancement 77 System) ………

Line 100 – suggestion: better bone healing

2. Materials and Methods

I think the division is confused.

I made a suggestion.

2.1. Dog selection and groups

All the dog owners of this study were clients of the Veterinary teaching Hospital of the University of Leon (Spain). A total of 30 dogs with unilateral CCLT were used. Body weight and age ranged between 5 and 65 kg and between 12 and 210 months, respectively. The animals were randomly distributed into two groups of fifteen dogs by gender, breed and affected side (Additional 123 file, Tables S1-S3):

Group 1- TTA with the implant Securos ® (model XGEN)

Group 2 - TTA with the implant Porous®

The inclusion criteria comprised the absence of any concurrent systemic or….

            A comprehensive clinical evaluation encompassing physical, neurologic and… The physical examination.... For hind limb examination.... in the stifle) [18].

2.1. Surgical procedures

            The surgeries were performed by two experienced surgeons. As we had two groups of fifteen animals, one surgeon did seven surgeries using….

Line 166 to line 222

2.3. Radiographic evaluation

            For OA assessment, radiographs were taken in all dogs the day of 134 surgery, one month and three m……

…..as good OI (Additional file, Table S18).

Line 134 to line 164

2.4. Experimental design

Line 197 - suggestion: Figures 1a and 1b

Line 219 - suggestion: Figures 2a and 2b

Line 279 - (43,26 Vs 43,60 minutes) – Please change: (43.26 Vs 43.60 minutes

Line 399 Please correct the formation: …..Vet Sci 2022;

Line 401   Please bold: 2003

Line 410         Please correct the formation: …..Schweiz Arch Tierheilkd. 2020, 162

Author Response

Good evening. Thank you very much again for your time. We really appreciate it.

We have read your comments and agreed to modify our manuscript according to your suggestions. All new changes are marked with a red color.

1.Introduction

Line 71: We have included both parts together in the same paragraph.

Line 100: we changed the sentence and wrote "better bone healing"

2. Materials and Methods

We agree that our division may be a little bit confusing. We have changed it and put it like you suggested:

2.1 Dog selection and groups : line 106

2.2 Surgical porcedures: line 134

2.3 Postoperative assessment: line 204

2.4 Experimental design: line 237

Thank you very much!

Line 197 Figures 1a and 1b : changed new line 165

Line 219 Figures 2a and 2b : changed new line 195

Line 279 43.26 vs 43.60 : changed new line 293

Bibliography

We corrected the formation in line 399 and line 410, and bold 2003 in line 401.

Thank you very much once more time and have a great day.

Kind regards.